# Eating Advice for People Who Wear Dentures: A Scoping Review

**DOI:** 10.3390/ijerph19148846

**Published:** 2022-07-21

**Authors:** Paula Moynihan, Roshan Varghese

**Affiliations:** 1Adelaide Dental School, Faculty of Health and Medical Sciences, The University of Adelaide, Adelaide, SA 5005, Australia; 2Glaxosmithkline Consumer Healthcare, Weybridge K13 0DE, UK; roshan.x.varghese@gsk.com

**Keywords:** nutrition, tooth loss, eating, dentures, dietary advice

## Abstract

Objective: A scoping review of available advice to address eating problems experienced by people who wear dentures was conducted in accordance with the PRISMA statement. The objective was to identify and map type, volume, and content of the available eating advice. Methods: Medline, CINAHL, and grey literature databases and Google were searched. Relevant content pertaining to study type, peer-review vs. grey literature, country of origin, advice content, and methods to evaluate effectiveness was mapped. Results: Of the 4591 records identified from peer-reviewed literature, 56 full papers underwent duplicate screening, resulting in 26 papers (from Germany (*n* = 1), Europe (*n* = 1), India (*n* = 2), Japan (*n* = 7), UK (*n* = 6), USA (*n* = 6), or other (*n* = 3)) being included in the review. These yielded 18 different items of relevant eating advice. Of the 258 screened websites, 63 were included, yielding 30 different items of eating advice. The most-cited advice was to eat soft food and avoid hard and sticky food, cut food into bite-sized pieces, and chew on both sides of the mouth and chew slowly and thoroughly. The identified advice was not supported by reference to peer-reviewed evidence. Advice included some conflicting messages and some advice was non-compliant with authoritative nutritional advice (e.g., avoid red meat, take a vitamin supplement). Conclusion: There is support for providing eating advice at the time of denture provision. A broad range of advice based on clinical experience to support people who wear dentures to overcome the functional limitations exists. However, the efficacy of this advice in improving diet and eating experience has not been tested.

## 1. Introduction

Significant tooth loss is associated with consumption of a diet lower in fruits and vegetables [1,2], dietary fibre [3], and protein [3,4]. The provision of dental prostheses (dentures) in the absence of dietary intervention does not result in consumption of a healthier diet [5,6,7]. However, evidence synthesis indicates that healthier eating intervention delivered concurrently with the provision of dental prostheses improves nutritional intake [8], thus indicating the efficaciousness of dietary intervention in conjunction with denture provision. 

Provision of dentures improves chewing and biting function but not to the extent of a dentate person [9]. People that wear dentures report problems with biting, chewing, pain on eating, denture-related stomatitis, sore mouth, reduced flavour perception, embarrassment caused by foods sticking to dentures, and limitations in food choice [10,11]. However, most research pertaining to dietary intervention in people who wear dentures has focused on improving nutritional intake as opposed to specifically addressing the practical and functional issues around eating that people who use dentures face. Indeed, there has been no review to scope the extent of eating advice for people who use dentures, including the source of the advice, the content of the advice, and any data on the effectiveness of advice at overcoming eating problems, improving nutrition, and supporting eating-related quality of life. In view of this, the overall aim of this study was to scope and critically appraise the body of information pertaining to patient advice that specifically addresses eating problems caused by wearing dentures. The overall review question was ‘What dietary or eating advice or interventions exist that address improving eating with dentures?’ The specific aims were (a) to identify the types of health education and intervention information available and whether the advice is evidence-based or based on clinical experience, (b) to note if the effectiveness of dietary or eating advice has been tested, and (c) to summarise the implications of the findings for the provision of evidence-based advice for improving eating experience with dentures in dental practice. The objective was to carry out a scoping review to identify and map existing information on dietary or eating advice to address eating with dentures regarding (1) types and volume of information, (2) geographical location (country), (3) place of intervention delivery (e.g., dental practice, general health care, online advice), (4) key content of advice, (5) whether advice is evidence-based or based on clinical experience, and (6) means of evaluation.

## 2. Materials and Methods

The detailed protocol for the scoping review is available on Figshare [12].

### 2.1. Population, Context and Concept (PCC)

Included in the review were publications pertaining to health education information on dietary advice and intervention to support eating ability targeted at adults aged 18 years and over who wear dentures (conventional dentures or implant-supported dentures, and complete dentures or partial dentures). Information from any country and that aimed at participants new to wearing dentures as well as people who were experienced in wearing dentures for any duration was included. Information and or interventions that were targeted at patients in the acute hospital setting and people living in residential care where they would not have control over their food provision and or preparation were excluded.

The concept being explored was eating or dietary advice and interventions specifically targeted at people who wear dentures, including one-to-one counselling and the provision of information through written materials and websites. This included food-based interventions aimed at overcoming eating problems inherent to wearing dentures and or promoting a healthier diet. Excluded were interventions pertaining to dietary therapy for specific diseases and the provision of nutritional supplements or enteral and parenteral nutritional support.

The context was healthcare settings (non-acute), in any country and online website from Australia, Canada, New Zealand, United Kingdom, and the US providing health education information on the topic.

### 2.2. Types of Information

Included were peer-reviewed intervention studies (RCT, non-randomised trial, quasi-experimental studies) and reviews and opinion pieces that made reference to dietary or eating interventions) and grey literature (i.e., government guidelines, online advice from recognised healthcare professionals or health care companies and guidelines of healthcare and relevant professional organisations). Excluded were articles written by lay press (including online information). Articles were included if they possessed an abstract in English. Abstracts and pre-prints were excluded.

### 2.3. Search Strategy

The PCC was used to define initial search terms and development of search strings for both peer-reviewed databases and grey literature. An initial scoping search was performed on Medline via PubMed using MeSH thesaurus terms and/or common terms to determine the search sensitivity and the need for search term or concept synonyms and variations. This was followed by an analysis of the text words contained in the title and summary of retrieved articles, and of the index terms used to describe the articles. This provided an opportunity to adjust search string syntax to optimise search sensitivity. An optimised search string was then used to search Medline (via the PubMed interface) and CINAHL (Cumulative Index to Nursing and Allied Health Literature) to capture peer-reviewed articles. The final search string combined two sets of synonyms: one listing alternative words for dentures (denture” OR “removable prosthodontics” OR “dentition” OR “Edentate” OR “partially dentate” OR “edentulism” OR “edentulous” OR “removable partial denture”) and one listing words relating to diet or eating related quality of life (“nutrition” OR “nutritional” OR “diet” OR “soft diet” OR “food” OR “soft food” OR “healthy eating” OR “dietary” OR “dietary advice” OR “eating advice” OR “eating” OR “quality of life” OR “health-related quality of life” OR “QoL” OR “HRQoL” OR “patient-reported outcome” OR “PRO”). 

Grey literature databases initially included OpenGrey, GreyMatters, and WorldWideScience. WorldWideScience was searched using the same compound search as PubMed. In addition, where appropriate, each of these databases was the subject of targeted searches, with specific additional filters or alternative search strings to identify anticipated sources of information. These were initially directed at documenting identified relevant national government documents and communications of prominent learned dental societies, oral health product manufacturers, and other dental organisations. This process helped to identify which type of bodies develop this type of nutrition guidance and where the gaps in information provision exist. OpenGrey and GreyMatters were searched using the simplest search available: “denture”. However, as this generated no relevant hits, these databases were removed from the search strategy. 

A similar scoping search was performed on Google to optimise search terms and to determine search sensitivity. Google searches required the definition of a series of filtering criteria at the level of the search. Ultimately, the Google search strategy used the ‘advanced search’ function to identify sites that contained all five common language key terms: “denture”, “eat”, “diet”, “food”, and “advice”. As Google has no limitations on the number of hits, Google searches were restricted to a maximum of 100 results. Moreover, Google requires searches to identify a ‘region’ (which typically defaults to the country in which the search is performed). In order to get a broad geographic representation of relevant hits, the first 20 hits from the five principal English-speaking Google ‘regions’, Australia, Canada, New Zealand, the United Kingdom, and the United States, were included. Each was required to be unique; accordingly, any repeat hits were omitted and the next in the list selected (this search was termed ‘regional web search’).

Targeted learned societies (catering primarily to the prosthodontics speciality and addressing clinical and patient aspects around denture provision) for grey literature searches were identified: American College of Prosthodontists, American Dental Association, Authority Dental Organization, European Prosthodontic Association, FDI (Federation Dentaire Internationale), International Association Dental Research (IADR), International College of Prosthodontists, Mouth Healthy, and Oral Health Foundation. Targeted commercial organisations (as in consumer-facing companies that manufacture and address products designed for caring for denture wearer needs) were initially identified: Denture Living, Fitty Dent, My Denture Care, and Steradent. Target sites were searched for individual elements of the Google search string (“denture”, “eat”, “diet”, “food”, and “advice”) using the Google advanced search function to ‘search within site’ (this search was termed ‘professional association search’).

Single dental-practice-based content from Google searches that was consistent with the research question was also identified. Using Google, the first 20 dental practices for each of the five English-speaking Google regions were identified using the search term ‘dental practice’ to list 100 dental practices in total. These sites were then searched for the words “denture”, “eat”, “diet”, “food”, and “advice”, using the Google advanced search function to ‘search within site’ (this search was termed ‘dental practice search’).

The date limits were 2010 to April 2021 for peer-reviewed literature and other grey information to capture recent evidence, guidance, and health information on nutrition.

### 2.4. Screening

Titles and summaries of all records identified in the electronic search were reviewed by one reviewer (AD) to exclude articles obviously outside the scope of the review. A random 10% sample of titles and abstracts or summaries were screened in duplicate by a second reviewer (AP) and inter-rater reliability (IRR) was assessed. Any differences between the reviewers’ decisions were resolved by discussion. When the articles apparently met the inclusion criteria or when there was not enough information in the abstract or summary, two reviewers screened the full article. Any differences between the reviewers’ decisions were resolved by discussion and when consensus could not be reached, a third reviewer (PM) was consulted. The reasons for exclusion of articles at this phase was logged. The reference lists of identified reports and articles were also searched for additional studies. 

### 2.5. Extracting and Charting Results

Extracting and charting results was undertaken by one reviewer and checked by a second reviewer. Disagreements were resolved by consensus with involvement of a third reviewer where necessary. Extracted data included information on author and year, country, origin and setting of dietary information or intervention (i.e., dental practice, general healthcare, online advice), peer-reviewed versus grey literature, study design (e.g., RCT, non-randomised trial, quasi-experimental, quantitative, qualitative; if relevant), aims and purpose of article, study population and sample size (if relevant), mode of information or intervention, duration of the intervention (if relevant), key content (eating advice, nutrition, and or behavioural change techniques, evidence- or experience-based), intensity or frequency of delivery of intervention content, and how the outcome of the intervention was measured (if relevant). The charting of information was trialled on at least one peer-reviewed article and at least one website. The form for charting results was created in MS Excel and was modified in an iterative way throughout charting. Examples of the charting forms are provided in the Appendix A Table A1 and Table A2. The data extracted on the content of the eating or dietary advice was summarised by tabulation (Table 1 and Table 2).

## 3. Results

### 3.1. Search Results

Figure 1 presents the PRISMA (Preferred Reporting Items for Systematic Reviews and Meta-Analyses) flow chart. Of the 4518 hits identified from the Medline, CINAHL, and WorldWideScience databases, 56 were included for full screening and 4535 were clearly outside the scope of the review. Of the 56 full texts reviewed, 20 were duplicates, 10 did not contain relevant content and 26 were included. Of these, 13 papers reporting seven studies contained original data to address the review concept: one RCT and six quasi-experimental studies. Other peer review papers (*n* = 13) did not contain original data but referred to dietary interventions for people who wear dentures, narratively as part of a review or in a discussion of an original article.

Included articles were from Germany (*n* = 1), Europe (*n* = 1), India (*n* = 2), Japan (*n* = 7), UK (*n* = 6), and USA (*n* = 6), with a further 3 not being affiliated to a specific geographical location. For both screening stages (title and summary, and full text), the IRR (inter-rater reliability) was 100%. 

Figure 1 also provides details of the number of records identified from the Google searches of 100 regional websites (20 from each: Australia, Canada, New Zealand, UK, and US), the aforementioned professional associations, and 100 dental practices. The Google searches led to the identification of 258 relevant hits. Following an assessment of the content, 63 were relevant to the objectives of this review: 57 identified in the regional website search and 6 identified from the targeted search of professional associations. The search resulting in the first 20 hits for dental practices from each of the included regions yielded no relevant hits. Of the included websites, 41 were for dentists or denturist practices, 8 were health associations, 5 were dental supply manufacturers, 3 were health service providers, and 6 were of other origins (see Appendix A Table A3).

### 3.2. Findings from Peer Review Literature

Table 1 presents data from the peer-reviewed papers, including key content and the source of dietary advice, modes of advice, and where relevant, measures to evaluate the effectiveness of dietary intervention. Of the seven studies with original data, two studies based the dietary advice directly on government healthier eating guidelines, four employed a personalised approach to dietary intervention based on assessment of the patient’s diet and provision of tailored feedback, and one study simply referred to advising a ‘balanced diet’. No study with original data specifically referred to practical advice to overcome the functional limitations of eating with dentures. From the peer-reviewed literature, 18 distinct recommendations and tips for eating with dentures were identified: a summary of these recommendations is presented in Table 2. 

### 3.3. Findings from Google Searches

Identified data were classified into the organisation type, i.e., (1) health service provider; (2) health organisations; (3) dental supply manufacturers; (4) dental and denturist practices; and (5) other, e.g., information website providers. Within each category of organisation type, data were organised by country of origin (Australia, Canada, New Zealand, UK, and US). Examples of the relevant data extracted from the identified websites is provided in Appendix A Table A2 and website addresses are shown in Table A3. From these Google searches, a total of 30 distinct recommendations and tips for eating with dentures were identified: a summary of these recommendations is summarised in Table 3.

## 4. Discussion

An aim of this review was to identify the type and content of available health education and interventions for eating with dentures, and to note whether advice is evidence-based or based on clinical experience. The review identified seven studies testing dietary intervention but the primary focus of all studies was the promotion of healthier eating. The majority of the practical advice for eating with dentures identified in both the literature and Google searches appeared to be based on clinical experience. No references to the evaluation of the effectiveness of this advice in improving diet, intake of nutrients, and quality of life of people who wear dentures were provided.

As previously noted in a systematic review of interventions delivering contemporaneous dietary and dental intervention [8], the details of dietary interventions in the peer-reviewed literature are not well-described. Almost all of the identified practical advice for eating with dentures was provided by the narrative reviews (and websites) with only one original study referring to practical advice (removing seeds from recipes) as part of a healthier eating intervention [19]. However, most identified peer-reviewed literature, including data identified from systematic reviews [8,37], supported the concept of providing dietary advice to patients that wear dentures. Some of the literature recommended the involvement of physicians (to screen), referral to a dietitian for personalised advice, and referral to a speech and language therapist for advice on correct textural modification (e.g., to avoid choking and to avoid unnecessary adoption of a liquidised or ‘soft’ diet where not required). Dietitians are skilled in providing advice pertaining to the unique nutritional requirements of older adults and the factors that can impact on nutritional wellbeing, including tooth loss and wearing dentures. Indeed, the first dietary intervention study of denture wearers was designed and delivered by a nutritionist and dietitian [21]. Advice for eating with dentures is obviously an area that requires interdisciplinary collaboration.

### 4.1. Limitations of the Content of Available Eating Advice

Information on many of the identified websites included the suggestion that, after a period of adjusting to dentures, food choices can be the same as with natural teeth. However, no reference was made to support this suggestion. Based on what is known of the functional limitations of dentures on eating function [9], it is unlikely that those who wear dentures are able to maintain the diet they had when they had natural teeth, and such advice might give those who are new to dentures unrealistic expectations.

Several websites included dietary recommendations that did not fit with mainstream authoritative nutritional advice. For example, this includes the recommendation to use vitamin and mineral supplements (concerningly, in one instance, this came under a section on ‘healthier eating’). Overall, authoritative nutrition guidelines do not generally recommend the use of supplements for adults, except for vitamin D, and dietitians always advocate a ‘food first’ approach. Practical advice to overcome the limitations of eating with dentures should take precedence. Any use of supplements should be under the guidance of a qualified medical practitioner or dietitian. In several instances, the suggestion to avoid coffee due to a diuretic effect was made. This advice is, however, in conflict with advice from both Public Health England and the NHS, which both state that tea and coffee consumption make important contributions to overall fluid intake. Advice provided on websites was sometimes conflicting; for example, information on some sites advised avoiding chewing gum because it sticks to dentures whereas information on other sites recommended chewing gum to keep the mouth moist. Information on some websites recommended avoiding fruits with skins intact or those containing seeds, whilst information on other websites recommended eating seedless grapes and berries, both of which have skins. Information on most identified websites advised avoiding toasted bread and crackers, yet information on some websites advised to eat these foods with something to soften them, such as hummus or soup. Information on some websites advised substitutes for ‘difficult’ foods; however, in some instances the recommended swaps did not have a comparable nutritional profile, including, for example, a recommendation to eat grapes and berry fruits (low in protein) in place of nuts (a principal source of plant protein). Another concern was the identification of spurious claims regarding the impact of dentures on nutritional intake and status that were unsubstantiated. Moreover, websites used terms such as ‘proper nutrition’ and ‘healthy foods’ without explaining the meaning of these terms or providing examples of foods meeting these criteria.

In several cases, websites had directly reiterated advice of an authoritative body, e.g., the UK NHS (National Health Service). Though following such authoritative advice is to be commended, it is a pity that said advice was not more comprehensive (see Table A2). By contrast, some websites (e.g., Migneault F dental practice in Canada: www.francismigneault.com/denturist-services/nutrition/ accessed on 1 April 2021) provided comprehensive dietitian-composed advice on food selection.

### 4.2. Limitations of the Scoping Review

In this scoping review, the date limits were 1980 to the present date for peer-reviewed literature, and other searches were limited to 2010 onwards. Therefore, any data on eating advice prior to these dates were not identified. However, it was important to ensure that the data captured reflected relatively recent or current guidelines on nutrition. The study was limited to the regional Google search for the first 20 sites identified using the search terms “denture”, “eat”, “diet”, “food”, and “advice” in the advanced search function. The approach of limiting to the first 20 hits possibly led to missing some data; however, given the degree of overlap in information between sites, this is unlikely. This review included websites from English-speaking countries and therefore the results may not have captured all existing globally available online advice. Food culture varies widely and eating problems experienced by people who wear dentures is likely to vary due to cultural differences in food choice, food preparation, eating style, and environment. Broadening the web searches to include a greater diversity of countries would be desirable in providing insights into such cultural differences in approaches to supporting eating well for people who wear dentures. However, a broader web search that necessitated translation was beyond the resources and scope of the current study. Moreover, translation of information identified on non-English language sites may lead to loss of nuances in the eating advice during translation.

### 4.3. Future Research

There is a clear need for further research to evaluate the effectiveness—in improving diet and eating experience—of practical advice to support eating with dentures within the context of authoritative eating guidelines. This scoping review did not identify any eating advice for people living with dementia that wear dentures: there is clearly a need for more research in this area. Furthermore, the scope and content of advice on eating with dentures available for people from non-English speaking countries needs to be investigated. This would facilitate the efficacy of eating advice relevant to a diversity of food cultures to be evaluated. Moreover, this would promote increased understanding by clinicians of the need for cultural nuances in giving advice on eating with dentures.

## 5. Conclusions

Overall, the available literature suggests that there is support for providing eating advice to people at the time of denture provision as part of their total care; a broad range of advice offered to people who wear dentures has been identified. Most of the eating advice is aimed at overcoming the functional limitations of wearing dentures and is based on clinical experience, but not necessarily with the involvement of a dietitian. The efficacy and effectiveness of such advice on nutritional intake and eating-related quality of life has not been tested.

## Figures and Tables

**Figure 1 ijerph-19-08846-f001:**
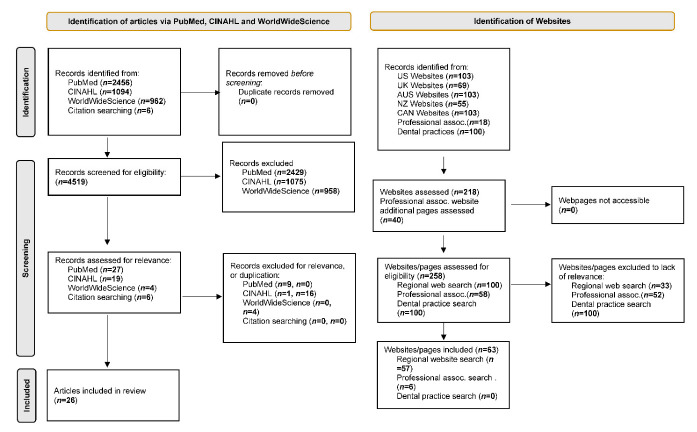
PRISMA flow chart.

**Table 1 ijerph-19-08846-t001:** Peer-reviewed papers with data and content relevant to the review.

Author/Year	Country	Diet Intervention/Advice	Study Population (*n*)	Objective	Evaluation Measures
Studies with primary date pertaining to dietary advice for patients with dentures					
Randomised controlled trials					
Komagamine et al., 2016 [13]; Amagai et al., 2017 [14]; Suzuki et al., 2018 [15]; 2019a [16]; 2019b [17]; Kanazawa et al., 2019 [18]	Japan	Dietary advice pamphlets (20 min to read) given on trial entry and day of dentures from two sources: Japanese Ministry of Agriculture, Forestry and Fisheries [Geriatric version] (Intervention) vs. American College of Prosthodontists [citing Felton et al. 2011 as control intervention.	Edentulous participants (70) randomised 1:1	Randomised controlled trial to investigate the combined effect of complete denture renewal and ‘simple’ dietary advice	OHIP-EDENT-J scores; Energy and nutrient intake; Masticatory function; MNA^®^-SF questionnaire; Antioxidant capacity; Anthropometry
**Quasi-experimental studies**					
Moynihan et al., 2012 [19] Ellis et al., 2010 [20] (based on Bradbury et al., 2006 [21]).	UK	2 × one-to-one customised dietary interventions delivered by a community nutrition assistant and the provision of an individually tailored nutrition education package to take home. Aimed to increase fruits, vegetables, and fibre (recipes for denture wearers, e.g., seed free were provided).	Patients wearing IODs or CDs aged 40–80 years (54)	Two-cohort prospective parallel dietary intervention study to identify and describe dietary strategies appropriate for dental patients who receive dental care that includes the placement and maintenance of dental prostheses	Dietary intake, Serum antioxidant status; Chewing satisfaction
Prakash et al., 2012 [22]	India	Participants were explained the importance of a well-balanced diet and the benefits of regular nutritional status assessment.	Edentulous patients aged ≥50–80 years in need of complete dentures for the 1st time. (94)	Single-armed study to evaluate the effect of edentulousness and prosthetic treatment on the nutritional status of the elderly population	MNA questionnaire, anthropometric measurements, diet assessment
Bartlett et al., 2013 [23]	UK	Two printed pamphlets providing information on healthy eating based on Eat well: a guide to healthy eating: Food Standards Agency UK 2005 and The good life. Food Standards Agency UK; 2007 (no denture-specific advice).	Edentulous patients attending Guy’s Hospital with complete dentures (35)	To investigate how nutritional advice and denture adhesives may be associated with eating healthier foods	Dietary intake (Health Equality Audit questionnaire). Eating ability (questionnaire from UK NDNS survey) OHIP Edent
Wöstmann et al., 2016 [24]	Germany	Individually tailored nutritional counselling was performed by a dietitian based on assessment of dietary intake and behaviour conducted in advance of the session.	Patients < 10 pairs of opposing natural teeth (25)	Intervention study to investigate the impact of implant-supported dentures and nutritional counselling on the nutritional status	3-day dietary record; MNA and BMI (nutritional status); Dental status including chewing ability; OHIP-G14,
Nabeshim et al., 2018 [25]	Japan	2 × 20-min basic dietary counselling sessions by a nutritionist, aimed at increasing fruit and vegetable intake and improving dietary habits with feedback and advice via a custom-made leaflet in accordance with Bradbury et al. (2006).	Patients (38) scheduled to receive RPDs, and currently eating < 350 g vegetables/day	Single-armed study to investigate the effects of dietary intervention on nutritional status in partially dentate patients receiving RPDs	Dietary intake (assessed by questionnaire) Serum vitamin C, α- and β-carotene); Masticatory function.
McCrum et al., 2020 [26]	UK	4 × meetings with a trained researcher to target habit-formation around three dietary domains (fruit/vegetables, wholegrains, healthy proteins).	Partially dentate older patients (57)	Feasibility study to develop and test a habit-based tailored dietary intervention, in conjunction with oral rehabilitation amongst partially dentate older adults	SRBAI, MNA questionnaires; anthropometry
**Articles with narrative information pertaining to eating with dentures (no primary data)**					
Shigli and Hebbal 2012 [27]	India	Authors recommend that dietary guidance, based on an assessment of the patient’s nutrition and dietary history, should be an integral part of prosthodontic treatment. As denture fabrication requires a series of appointments, dietary analysis and counselling can easily be incorporated into an edentulous patient’s treatment plan.	Patients undergoing complete denture procedure (35)	Single-armed study to assess these changes before and 1 month after placement of complete dentures in a dental hospital in Indore, Madhya Pradesh, India	10-item closed-ended questionnaire
Matloff, 2013 [28]	US	Author includes tips on how to determine the safest texture for edentulous patients. Including cognition and motivation to eat, fatigue, positioning (sit at 90 degrees), not speaking while eating. Involvement of a speech and language therapist to assess patient.	N/A	N/A	N/A
Touger-Decker et al., 2013 [29]	US	Advises cutting food into smaller sizes for biting and chewing ease. Moistening tough-to-chew foods. Dietary intervention should consider patient’s systemic diseases and disabilities and associated oral manifestations.	N/A	N/A	N/A
Mobley and Dounis 2013 [30]	US	Placement of a removable prosthesis should include a discussion involving patient-focused dietary strategies to promote healthful food choices, thus assisting patients to maintain optimum health. A first step is to identify a patient’s food-avoidance and modification behaviours. Then, practitioners can provide guidelines for adopting food selection and preparation methods to ensure an adequate diet. Instructions for modifying foods to support healthful dietary habits and food choices can lead to positive dietary consequences and ensure patient-centred care.	N/A	To identify and describe dietary strategies appropriate for dental patients who receive dental care that includes the placement and maintenance of dental prostheses	N/A
Yoshida et al., 2014 [31]	N/A	Dietary instruction by a registered dietitian after denture treatment is reported to be essential (citing Allen 2005; Bradbury et al., 2006).	N/A	N/A	N/A
Rathee et al., 2015 [32]	N/A	Prosthetic reconstruction can require a series of appointments, dietary analysis and counselling can be incorporated into the treatment sequence. In the first few days after insertion of denture, swallowing should be practiced, and a liquid diet prescribed. ‘Soft foods’ is advocated for the next few days, then a regular diet by the end of the week (citing Adams 1961; Detroit 1960).	N/A	N/A	N/A
Zelig et al., 2016 [33]	N/A	Discussion states: Several included studies proposed that nutrition support should be provided to individuals who wear dental prostheses. Oral health professionals can address diet and nutrition as part of their treatment plan by providing basic diet education or referring patients to registered dietitian. Nutrition intervention should be tailored to dental status and ability to chew and swallow.	N/A	N/A	N/A
Larson 2017 [34]	US	Harder or more solid foods, such as pieces of meat or harder pieces of vegetable’ and ‘alcohol consumption’ increases the risk of food impaction.	N/A	N/A	N/A
Zelig et al., 2018 [35]	US	Discussion states: the use of nutritional screening tools by oral healthcare professionals could help to provide timely referrals to primary care physicians or Registered Dietitians/Nutritionists. Referrals to community assistance programs (such as Meals on Wheels) could also be made as appropriate to prevent decline in nutrition status. No specific reference to wearing dentures made.	Patients aged ≥ 65 who attended an urban northeast US dental school clinic (107)	To explore the associations between nutritional and dentition status in older adults	MNA, MNA-SF, Self-MNA
Kossioni et al., 2018 [36]	Europe	Physicians need, appropriate teaching at both undergraduate and postgraduate levels and through continuing education courses to enable them to provide dietary advice. The physician should offer all patients advice on healthy and unhealthy dietary habits. Older people with dentures often face chewing difficulties-patients and caregivers should be advised on appropriate food selection and preparation, including cutting it into small portions, chopping, mashing, or moistening before chewing.	N/A	To describe practice guidelines and tools for physicians for promoting oral health in frail older adults, based on the competencies previously described in European recommendations	N/A
Al-Sultani et al., 2018 [12]	UK	Qualitative data from patients receiving new dentures showed eating problems with dentures persisted following new dentures (e.g., need to avoid foods with seeds and nuts when using fixative due to food entrapment; need to slice or stew apples; avoidance of sticky food when using fixative as pull off denture) indicating that eating advice to overcome these obstacles is warranted.	Complete denture wearers undergoing denture replacement (35)	To assess eating related quality of life before and after insertion of replacement complete dentures	ESIRE questionnaire (Kelly et al., 2012 [11]) ESIRE Score and qualitative data (verbatim quotes from patients).
Zelig et al., 2020 [37]	US	Discussion suggests all health professionals can screen for tooth loss and lack of occlusion through oral examination during annual physicals or well visits and can refer accordingly to oral health professionals and credentialed dietitians. Dietitians can ask about oral factors affecting one’s ability to consume foods and fluids; these findings can be integrated into care plans to reduce risk of malnutrition.	N/A	To conduct evidence synthesis addressing the question: Among adults aged ≥ 60 years living in developed countries, what are the associations between tooth loss and nutritional status as assessed by a validated nutrition screening or assessment tool?	N/A
McGowan et al., 2020 [8]	UK	Discussion concluded: Few interventions were theory-based, and intervention components were poorly described. Overall, narrative synthesis indicated support for dietary intervention coupled with oral rehabilitation on diet.	N/A	To synthesize literature relating to oral rehabilitation coupled with dietary intervention in adults.	N/A

BMI = body mass index. CDs = complete dentures. ESIRE = emotional and social issues relating to eating. IOD = implant overdenture. MNA = mini nutritional assessment tool. MNA SF = mini nutritional assessment tool short form. Self-MNA = self-completed mini nutritional assessment tool. NDNS = National Diet and Nutrition Survey. OHIP = oral health impact profile questionnaire. OHIP Edent = oral health impact profile questionnaire for edentulous populations. Edent G-14 = short form version of OHIP. SRBAI = Self-Report Behavioural Automaticity Index.

**Table 2 ijerph-19-08846-t002:** Summary of recommendations for dietary intervention in people who wear dentures, identified from the peer-reviewed literature.

Recommendation for Dietary Intervention	Supporting Citations
Moistening food to ease chewing.	[29,36]
Cut food into smaller pieces to ease biting.	[29]
Mash food.	[36]
Modify recipes to make them ‘seed free’.	[19]
Identify patient’s food avoidances and modifications to food preparation and integrate healthier eating advice around limitations.	[30,33]
Sit upright when eating.	[28]
Avoid speaking with a food bolus in mouth.	[28]
Provide advice on adopting food selection (no specific example provided).	[30,36]
Provide advice on how to modify food preparation (no specific examples provided).	[30,36]
Eat liquid food for first few days, followed by ‘soft diet’ for a few days then wean to regular diet by one week (for patients new to dentures).	[32]
Advise that hard foods (meat pieces, hard vegetables) and consuming alcohol can increase risk of food impaction (choking).	[34]
Avoid sticky foods as pull denture away from fixative.	[12]
Avoid foods with seeds and nuts that can get trapped under denture especially if using fixative.	[12]
Slice apples or stew.	[12]
Practitioners should use a nutrition screening tool and refer those at nutritional risk to a registered dietitian.	[35]
Integrate a series of personalised dietary intervention with appointments for denture fabrication.	[27,33]
Seek expertise of speech and language therapist for correct level of textural modification.	[28]
Seek advice of dietitian (who should enquire about chewing ability).	[31,33,37]

**Table 3 ijerph-19-08846-t003:** Summary of recommendations for dietary intervention in people who wear dentures, identified from the Google searches.

Advice	No of Mentions	Specific Details and or Contraindications
Eat soft foods (easy to chew).	38	Mostly with reference to the early stages post insertion
Cut food into tiny pieces/bite sized pieces.	34	Tiny pieces of food can cause choking; bite-sized is correct
Avoid tough, crunchy, or hard food.	32	Especially in early stages
Chew on both sides (back teeth) to keep dentures more stable while you eat.	29	Avoiding small food items that tend to be eaten on one side was also advised
Avoid sticky foods.	28	Dried fruit, peanut butter, chewing gum were examples
Chew/eat slowly (thoroughly).	27	Especially, to avoid risk of choking
Avoid hot foods; denture insulating can risk burning your mouth.	20	Test temperature on lip
You will be able to eat your normal diet.	16	Gives a false sense of hope as unlikely to restore to full dentate function
Avoid foods with sharp/hard edges.	14	e.g., crusty bread, nuts
Start with liquids, purees (use a blender, smoothies) liquid meal replacements.	12	No mention of degree of textural modification
Take small bites/mouthfuls.	10	
Avoid (tough) (red) meat or ‘chewy foods’.	9	Only tough red meat needs to be avoided
Take it slowly; gradually progress from soft to harder foods.	9	Risk denture damage Risk of choking
Avoid seeds (tiny bits) as they get stuck beneath denture.	9	Possible alternative would be to rinse mouth after eating seeds
Adhesives can help (stop food getting trapped/help cope with harder foods).	8	
Drink water (liquid) while eating.	8	To stops starchy foods sticking, to ease chewing/swallowing)
Bite using canines not incisors.	8	
Don’t use toothpicks.	7	
You will need to prepare food differently.	7	
Stewed or slow-cooked meats.	7	
Take a multivitamin and mineral supplement.	5	Most authoritative advice recommends only vitamin D
Avoid spicy foods.	4	Relevant for sore mouths
Avoid coffee due to diuretic.	3	Authoritative advice includes coffee as part of fluid intake
Speak to a dietitian (or nutritionist).	3	
Eat a balanced diet (‘proper nutrition’).	3	No indication of what constitutes a balanced diet was provided
Moisten foods.	2	
Taste [flavour] can be diminished.	2	
Avoid foods you need to bite into (pizza).	1	
Avoid mints with eucalyptus oil as can dissolve acrylic dentures.	1	
If problems with eating (after time) see your dentist.	1	

## Data Availability

Data are contained within the article and Appendix. The protocol for the review is available on Figshare [9]. The URLs to the websites identified in the searches are provided in the article (Appendix A Table A3).

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
