# Peer review of "Eating Advice for People Who Wear Dentures: A Scoping Review"

_ijerph, 2022, doi:10.3390/ijerph19148846_

Round 1
Reviewer 1 Report
I thought it was a very unique research.
In the increase of elderly people, this paper provides very important information.
On the other hand, I have some questions.
Although the text suggests the importance of dietitians' involvement, please cite a paper on whether dietitians are educated in dealing with denture patients.
In addition, I would like you to present and consider cases that actually accompany nutritional disorders due to lack of guidance.
Finally, what kind of guidance is appropriate for people with dementia?
Please revise "Future Research" and "Conclusion" including the above.
Reviewer 2 Report
Dear Authors, thank you for your work, but on the end, you didn't find anything new in your work. I don't understand why you include the recommendations of dental practitioners, which basically are on base on clinical experience not on EBM.
Reviewer 3 Report
The authors aimed to carry out a scoping review to identify and map existing information on dietary/eating advice to address eating with dentures regarding: (1) types and volume of information; (2) geographical location (country); (3) place of intervention delivery (e.g., dental practice, general health care, online advice); (4) key content of advice (5) whether advice is evidence-based or based on clinical experience and (6) means of evaluation.
The study covers some issues that have been overlooked in other similar topics. The structure of the manuscript appears adequate and well divided in the sections. Moreover, the study is easy to follow, but some issues should be improved. Some of the comments that would improve the overall quality of the study are:
a. Authors must pay attention to the technical terms acronyms they used in the text.
b. English language needs to be revised.
c. Conclusion Section: This paragraph required a general revision to eliminate redundant sentences and to add some "take-home message".
Round 2
Reviewer 2 Report
I'm sorry the response of the Authors is not enough
Author Response
The reviewers comment is somewhat vague and they provide no indication of what they would like changed.
We re-iterate that the purpose of a scoping review is to identify the volume and nature of information published (in both peer review and grey literature) on a subject. The purpose was to identify and map what dietary advice exists for patients who wear denture irrespective of whether it was evidence based or based on clinical experience. The purpose was not to make recommendations on what dietary advice to give. What this scoping review has shown is that there is a considerable volume of information that patients can access but the efficacy of this advice has not been tested - that is the important finding of this scoping review (which we feel we have made very clear). If the reviewer would like us to make amendments to the paper then we would welcome specific comments to which we can respond. However, we are unable to change the purpose of our research which was to scope what information exists on advice for eating with dentures.
I respect that reviewers can have an opinion but I challenge the scores that are marked below (which seem to have been amended from the original scores despite the manuscript largely remaining the same). The work has been conducted as a scoping review under the robust guidelines of the JBI Institute and the protocol was independent reviewed by experts in the methodology. The manuscript has been written by two experienced authors whose first language is English.